# Mantle Cell Lymphoma Presenting as Acute Abdominal Syndrome: A Rare Case Report and Literature Review

**DOI:** 10.3390/healthcare9081000

**Published:** 2021-08-05

**Authors:** Fu-Chou Lee, Junn-Liang Chang, Hung-Ming Chen, Wan-Chen Tsai, Po-Jen Hsiao

**Affiliations:** 1Department of Surgery, Taoyuan Armed Forces General Hospital, Taoyuan 325, Taiwan; colin155645@gmail.com; 2Department of Pathology and Laboratory Medicine, Taoyuan Armed Forces General Hospital, Taoyuan 325, Taiwan; junn9liang@yahoo.com.tw; 3Division of Hematology and Oncology, Department of Internal Medicine, Taoyuan Armed Forces General Hospital, Taoyuan 325, Taiwan; mptt@aftygh.gov.tw; 4Division of Hematology and Oncology, Department of Internal Medicine, Tri-Service General Hospital, National Defense Medical Center, Taipei 114, Taiwan; 5Division of General Surgery, Department of Surgery, Taoyuan Armed Forces General Hospital, Taoyuan 325, Taiwan; greenmai2013@gmail.com; 6Division of General Surgery, Department of Surgery, Tri-Service General Hospital, National Defense Medical Center, Taipei 114, Taiwan; 7Division of Nephrology, Department of Internal Medicine, Tri-Service General Hospital, National Defense Medical Center, Taipei 114, Taiwan; 8Division of Nephrology, Department of Internal Medicine, Taoyuan Armed Forces General Hospital, Taoyuan 325, Taiwan; 9Department of Life Sciences, National Central University, Taoyuan 325, Taiwan; 10Big Data Research Center, Fu-Jen Catholic University, New Taipei City 242, Taiwan

**Keywords:** mantle cell lymphoma, non-Hodgkin’s lymphoma, immunochemical biomarkers, cyclin D1, SOX-11, acute abdominal syndrome, abdominal pain, appendicitis, appendix

## Abstract

Background: Acute abdominal syndrome can be caused by several possible reasons. The most common causes are perforation of a gastroduodenal ulcer, peritonitis, intestinal obstructions, and perforation of an appendix or fallopian tube. Fever and pain can be caused by an appendicitis or sigmoiditis. Appendiceal lymphoma is a rare disease that is usually found incidentally during appendectomy. Most of the cases are non-Hodgkin’s lymphomas. Mantle cell lymphoma is an aggressive B-cell non-Hodgkin’s lymphoma with a poorer prognosis than other B-cell lymphomas; thus, a definitive diagnosis is essential. Case Summary: A 60-year-old man presented with right lower quadrant pain. He denied any nausea, vomiting or anorexia and was afebrile. The physical examination revealed right lower quadrant abdomen tenderness. The computed tomography scan revealed periappendiceal fatty stranding with a swollen appendix, approximately 2 cm in diameter and prominent paraaortic, portacaval and mesenteric lymph nodes. A diagnosis of acute appendicitis was made, and laparoscopic appendectomy was performed immediately. The subsequent pathological examination revealed severe congestion with lymphoid hyperplasia. The immunohistochemistry stains revealed positive staining for cluster of differentiation (CD) CD20, B-cell lymphoma-2 (Bcl-2), cyclin D1, SRY-box transcription factor-11 (SOX-11), immunoglobulin D (IgD) and immunoglobulin M (IgM) but negative staining for CD3, CD5, CD10 and CD23. 18F-FDG positron emission tomography showed peripheral lymph node involvement, while the bone marrow biopsy showed negative findings. Therefore, a diagnosis of mantle cell lymphoma, Ann Arbor stage IVA, was made. The patient received postoperative combination chemotherapy and remained in a stable condition over a 1-year follow-up period. Conclusion: We report an uncommon case that initially presented as acute appendicitis, for which a final diagnosis of mantle cell lymphoma was made. In comparison with other B-cell lymphomas, mantle cell lymphoma has a poorer prognosis, and positive immunochemical staining of cyclin D1 and SOX-11 is useful for differentiating mantle cell lymphoma from other appendiceal lymphomas and treating patients appropriately. Physicians and nursing staff should be also aware of the associated complications and management in these patients.

## 1. Introduction

Acute abdominal syndrome can present with a variety of clinical conditions from benign and self-limited disease to surgical emergencies. Evaluating abdominal pain needs an approach based on likelihood of the disease, medical history, physical examination, laboratory studies, and further imaging investigations. Location of the pain is a valuable preliminary point and can provide a guide for further evaluation. For example, acute right lower quadrant pain in the patients may strongly propose acute appendicitis. It is also imperative to consider special populations such as women, who may present with genitourinary diseases and the elderly, who are at risk of malignancies. Approximately 1% of the appendectomy specimens are found to be appendiceal neoplasms. Most of these tumors are neuroendocrine tumors and epithelial tumors. Other tumors are rarely encountered, including lymphoma, metastases, mesenchymal tumors, sarcoma and neuroectodermal and nerve sheath tumors [1]. The gastrointestinal tract is the most common extranodal site involved in lymphomas, accounting for 5–20% of all cases [2]. The most commonly involved site is the stomach, followed by the small bowel, terminal ileum and cecum region [3].

Mantle cell lymphoma (MCL) is a rare and generally aggressive B-cell non-Hodgkin’s lymphoma (NHL), accounting for 2–4% of all NHL cases [4,5]. MCL typically initially present as lymphadenopathy, and up to one-third of patients have systemic B symptoms and frequently have disease involvement in extranodal sites. Common sites of extranodal involvement include the gastrointestinal tract, breast, pleura, and orbit [6]. Gastrointestinal involvement most commonly occurs in the colon and stomach [7]. Compared with other B-cell NHLs, MCL has earlier relapse and poorer long-term survival [8]. Staining for cyclin D1 and SRY-box transcription factor-11 (SOX-11) with immunohistochemistry are useful markers to reach a definitive diagnosis of MCL. MCL involving the appendix complicated by acute appendicitis has rarely been described previously. Herein, we report a case of mantle cell lymphoma that initially presented as acute appendicitis, emphasizing the need to consider differential diagnoses with appropriate healthcare and management. In this manuscript, a literature search was performed in PubMed and Scopus for all publications from inception published before 31 October 2020. Studies reporting cases of MCL were included. Basic science articles, editorials and correspondence were also reviewed.

## 2. Case Presentation

A 60-year-old man presented to our emergency department with right lower quadrant abdominal pain for 1 day. He denied any nausea, vomiting or anorexia. He had a history of hypertension, gouty arthritis, gall stones and renal stones. On initial assessment, the patient was afebrile and had a high blood pressure (153/104 mmHg) and normal heartbeat (75 bpm). The physical examination showed mild tenderness in the right lower quadrant of the abdomen, but the abdomen was soft with no muscle guarding. The laboratory findings revealed leukocytosis (white blood cell count, 18,270 cells/mm^3^); serum electrolytes and renal and liver function tests were within normal limits (Table 1). The urinalysis and Bence-Jones protein results were unremarkable. The abdomen computed tomography scan showed periappendiceal fatty stranding with a swollen appendix (Figure 1) and prominent paraaortic, portacaval and mesenteric lymph nodes. A diagnosis of acute appendicitis was made, and laparoscopic appendectomy (Appendix A) was performed immediately.

In the pathological assessment, the appendix was measured to be 9 cm in length and 2 cm in maximal diameter, and the lumen was severely congested and filled with fecalith. Microscopically, marked reactive atypical cell hyperplasia with mild polymorphism in the mantle zones, variety of cell types of naive pregerminal center cells, starry sky-like appearance with obscure hyalinized vascular proliferation and epithelioid histiocytes with lamina propria, submucosa, penetrating into muscularis layer, serosa and mesenteric fibroadipose were found. These tumor cells were characterized by a rather uniform and monotonous small to intermediate lymphocytic proliferation. These atypical cells demonstrated diffusely positive immunoreactivity for cluster of differentiation (CD) CD20, and B-cell lymphoma-2 (Bcl-2). They also presented positive immunostaining for cyclin D1 (over expression) of atypical lymphoid cells, and negative for pan Cytokeratin antibody (pan-CK), epithelial membrane antigen (EMA), S-100, CD10, and CD23. All of the above findings were consistent with malignant B-cell lymphoma, and the final diagnosis of CD5-negative MCL was made based on the histology features (Appendix A). The sections showed severe congestion with lymphoid hyperplasia (Figure 2A), characterized by small- to medium-sized monotonous atypical lymphoid cells (Figure 2B). The specimens revealed mantle cell lymphoma, with immunohistochemistry stains positive for CD20 (Figure 2C), Bcl-2, cyclin D1 (Figure 2D), SRY-box transcription factor-11 (SOX-11), immunoglobulin D (IgD) and immunoglobulin M (IgM) but negative for CD3, CD5, CD10 and CD23. The Ki-67 proliferation index was approximately 20%. Further examinations showed that 18F-FDG positron emission tomography showed peripheral lymph node enlargement, while the bone marrow biopsy showed no malignant cells. According to the Ann Arbor staging system, the final diagnosis was stage IVA mantle cell lymphoma. The patient received an adjuvant combination chemotherapy regimen and remained in a stable condition over a 1-year follow-up period.

## 3. Discussion

MCL is rare and generally aggressive B-cell NHL that usually occurs in middle-aged adults, has a male predominance and presents with advanced stage at diagnosis. It typically presents with generalized lymphadenopathy, and extranodal involvement is common, including bone marrow, peripheral blood, spleen, and Waldeyer’s ring involvement and invasion into the gastrointestinal tract, which may present as a distinctive symptom of multiple lymphomatous polyposis of the intestine. Most appendiceal lymphomas are non-Hodgkin’s lymphoma, and the most common histologic type is large B-cell lymphoma, followed by Burkitt lymphoma [9]. Although gastrointestinal lymphoma is the most common form of extranodal lymphoma, patients with mantle cell lymphoma of the appendix are rare, and only a few cases can be found in the literature (Table 2) [10,11,12,13,14,15]. Compared with other B-cell lymphomas, MCLs have poorer long-term survival and earlier relapse even with appropriate therapy. Thus, a definitive diagnosis is essential. The gold standard diagnostic procedure for MCL is histopathological examination and immunohistochemistry analysis. Morphologically, MCL generally comprises monomorphic small- to medium-sized lymphoid cells with irregular nuclear contours that infiltrate and expand to the mantle zone. Molecularly, MCL is characterized by the t(11;14)(q13;q32) translocation, which juxtaposes the cyclin D1 (CCND1) gene (also called B-cell lymphoma-1 (Bcl-1) gene) and the gene encoding the immunoglobulin heavy chain, leading to cyclin D1 overexpression [7,8,9,16,17].

Cyclin D1-positive human cancers include mantle cell lymphomas, breast cancers, head and neck carcinomas, oropharyngeal cancers, hepatocellular cancers, colorectal cancers, skin cancers and sarcomas [18]. Cyclin D1 and D3 play a role in the regulation of the G1 to S phase transition of the cell cycle, acting via phosphorylation of the retinoblastoma gene product [19,20]. The translocation can be identified by immunochemistry, fluorescence in situ hybridization or polymerase chain reaction. A few cases of cyclin D1-negative MCL have been reported, and these cases typically lack evidence of chromosome translocations or genomic amplifications but are positive for cyclin D2 or D3 [21]. For cyclin D1-negative MCL, SOX-11 is a useful biomarker and helps distinguish indolent SOX-11-negative MCL from classical MCL [22,23]. Many publications have indicated that a lack of SOX-11 expression in MCL patients is a feature of a nonaggressive clinical course [24]. MCLs usually express pan-B-cell antigens (e.g., CD19 and CD20), CD5 and cyclin D1 but are negative for CD10 and CD23; only 5% of MCL patients lack expression of CD5 and cyclin D1 [5,6,7,8,9,17,25,26]. CD5 is a marker of T cells that is not typically expressed on B-cells. A retrospective review concluded that CD5-negative MCL had a more favorable prognosis than CD5-positive MCL [26,27,28]. Biomarkers, including CD5, cyclin D1 and SOX-11, are useful for the differential diagnosis of mantle cell lymphoma from other B-cell NHLs. Atypical chronic lymphocytic leukemia (CLL) and small lymphocytic lymphoma (SLL) may also demonstrate positive for cyclin D1 but do not have CCND1 translocations and usually lack SOX-11 expression [29]. Rare cases of diffuse large B-cell lymphomas could be positive for cyclin D1, but they frequently lack CCND1 gene translocation and lack immunoreactivity for SOX-11 [30]. Hairy cell leukemia (HCL) may show positive for cyclin D1 and initially be misdiagnosed as MCL. The absence of the CCND1 gene fusion and the strong expression of CD11C, CD25 and CD103 could be found in HCL [31]. MCL and multiple myeloma/plasmacytoma with 11q13 abnormalities typically express Cyclin D1 transcript variants with the longer 5′-untranslated regions (5′ UTRs) [32].

Patients with gastrointestinal involvement by MCL can be identified by esophagogastroduodenoscopy and colonoscopy, with the majority of involvement being seen in the stomach (74.3%) or colon (57.1%) [33,34,35,36]. The endoscopic features varied morphologically; superficial types, protruded types, fold-thickening types, and ulcerative types were mostly found in the stomach, and MLP was dominant from the duodenum to the rectum [36]. Tissue biopsy by endoscopic ultrasonography can help to find submucosal lesions and may improve the diagnosis rate [37]. The prognosis of mantle cell lymphoma can be estimated by the Mantle Cell Lymphoma International Prognostic Index (MIPI) and Ki-67 proliferation index (PI). The Ki-67 proliferation index is a strong independent prognostic factor with a 20% cut-off value regardless of bone marrow involvement and a 30% cut-off value with bone marrow involvement [38,39]. To date, there are no prospective data guiding observation in patients based on SOX-11 or tumor protein p53 (TP53) expression. However, these are additional variables that can be considered clinically to predict disease course and prognosis. A recent cohort study recommended that over 50% TP53 expression had a poorer prognosis, while SOX-11 expression was not a reliable prognostic marker. The reason may be owing to an under-representation of patients with non-nodal disease. Another explanation could also be considered due to a difference in methods to determine SOX-11 positivity, gene expression profiling and improved clinical outcomes with low SOX-11 expression [40]. The main treatment modalities are combination chemotherapy plus immunotherapy. The treatment strategies for MCL may be variable and dependent on the symptoms and patient fitness. Despite recent advances, MCL remains incurable and patients with high-risk disease have particularly poor outcomes. R-CHOP regimen (rituximab with cyclophosphamide, doxorubicin, vincristine and prednisone) is the main first line chemotherapy. New insights into the pathogenesis of MCL and future therapies including biological therapy, targeted therapy, radiation therapy and stem cell transplantations have been reported [8]. Surgical intervention is rarely needed except for when symptoms are complicated with bowel obstruction, uncontrollable bleeding or acute appendicitis [41]. Hypercalcemia is also considered as a biomarker for the aggressiveness of lymphoma or leukemia associated diseases [42]. In summary, the developments in MCL have begun to have improvements in life quality and overall prognosis. Further clinical study results will provide more evidence of durable remissions and acceptable long-term side effect profiles. The continued improvement in targeted molecular signaling inhibitors based on the biology of MCL is also a therapeutic approach for this disease, which may have improvements in quality of life and survival prognosis among these patients.

Appendiceal neoplasms are uncommon tumors of the gastrointestinal tract that may present with symptoms similar to appendicitis and are diagnosed by appendectomy specimens. The pathogenesis is due to the enlarged lymphoma in the lumen of the appendix or extrinsic compression on the appendix by lymphomatous polyposis [9,10]. If the enlarged appendix measures larger than 15 mm in diameter, an appendiceal tumor should be considered because the inflamed appendix rarely exceeds 15 mm [25]. In addition, if the enlarged appendix measures 30 mm in diameter or larger, appendiceal lymphoma should be taken into consideration [9,25]. Although the abdomen CT image appearance may not be pathognomonic, a markedly enlarged appendix and regional lymphadenopathy may be observed.

## 4. Conclusions

We report a teaching case of malignancies initially presenting as appendicitis. Our report highlights that the differential diagnoses of acute appendicitis include appendiceal tumors such as extranodal lymphomas. Biomarkers, including cyclin D1 and SOX-11, are highly specific and sensitive for MCL. Physicians and nursing staff should be also aware of the possibility of an appendiceal tumor mimicking appendicitis and provide appropriate healthcare and management in these patients.

## Figures and Tables

**Figure 1 healthcare-09-01000-f001:**
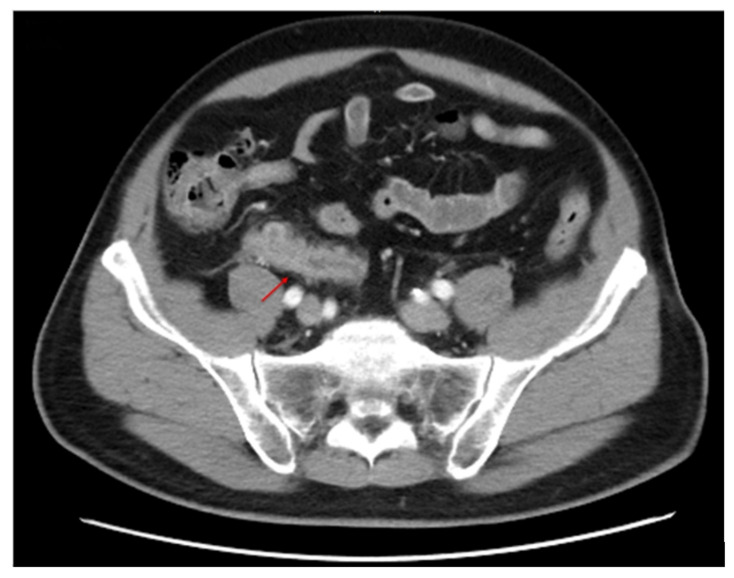
Periappendiceal fatty stranding and enlarged appendix (arrow).

**Figure 2 healthcare-09-01000-f002:**
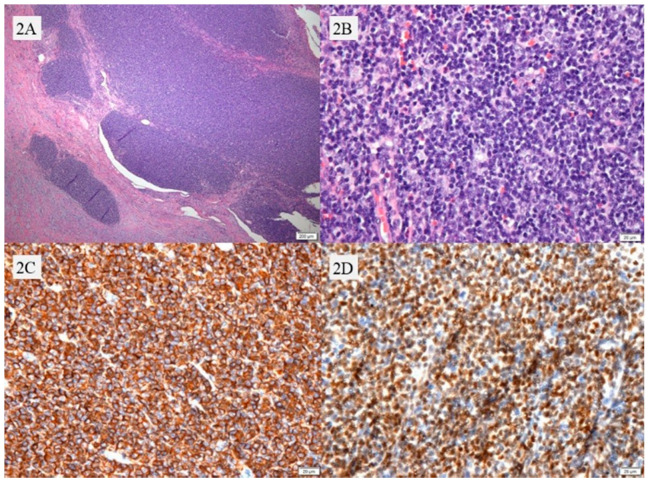
The hematoxylin and eosin (H&E) staining shows mantle cell lymphoma involving the appendiceal lumen with cellular infiltration of whole layers (**A**), original magnification ×4, characterized by small- to medium-sized monotonous atypical lymphoid cells (**B**), original magnification ×40. The neoplastic lymphoid cells showed diffuse positive immunohistochemical staining for CD20 (**C**), original magnification ×40 and Cyclin D1 (**D**), original magnification ×40.

**Table 1 healthcare-09-01000-t001:** Blood biochemistry data.

Parameters	Results	Normal Value
White blood cell count (/µL)	18,270	4800–10,800
Haemoglobin (g/dL)	15.2	12–16
Platelet count (/µL)	199,000	130,000–400,000
Mean corpuscular volume (fL)	86.7	81–99
BUN (mg/dL)	13.6	15–40
Creatinine (mg/dL)	0.99	0.9–1.8
Sodium (mEq/L)	140	133–145
Potassium (mEq/L)	4.11	3.8–5.0
Chloride (mEq/L)	97.4	96–106
Calcium (mg/dL)	9.1	8.5–10.5
Phosphate (mg/dL)	3.25	2.4–4.1
Uric acid (mg/dL)	7.2	1.9–7.5
GOT (IU/L)	24.8	11–47
GPT (IU/L)	30.3	7–53
Globulin (gm/dL)	2.48	1.4–3.5
Albumin (gm/dL)	3.28	3.5–5.5
A/G ratio	1.32	0.8–2.0
CRP (mg/dL)	0.198	<0.5
LDH (U/L)	386	120–240

BUN: blood urea nitrogen; GPT: glutamyl pyruvate transaminase; GOT: glutamyl oxaloacetic transaminase; A/G: albumin/globulin; CRP: C-reactive protein; LDH: lactate dehydrogenase.

**Table 2 healthcare-09-01000-t002:** Cases of mantle cell lymphoma with appendix involvement.

Reports	Age/Sex	Underlying Diseases	Surgical Intervention	CD Markers of Appendix	Clinical Manifestation	Survival Time
Tan et al. [10]	74/M	Gastric mantle cell lymphoma	Appendectomy	CD3(+), CD5(+), CD20(+), Cyclin D1(+) CD10(−) Ki-67:10–20%	Right lower abdomen pain	3.5 years
Rahimi et al. [11]	65/F	Peritoneal mantle cell lymphoma Gastrointestinal Stromal Tumor	Right-sided hemicolectomy	CD5(+), Pax-5(+), CD20(+), Cyclin D1(+), Bcl-2(+) CD10(−), CD23(−)	Symptom-free period of 5 years and then had recurrence. Abdomen CT revealed tumor over ileocecal region.	>2 years
Linden et al. [12]	71/M	Mantle cell lymphoma	Appendectomy	CD3(+), CD5(+), CD20(+), Cyclin D1(+) Bcl-6(−), Ki-67:60–80%	Right-sided abdominal after starting 1st cycle of chemotherapy. Abdomen CT revealed enlarged appendix with perforation.	6 months
Gaopande et al. [13]	50/F	Acute calculous cholecystitis	Cholecystectomy Appendectomy	CD20(+), cyclin D1(+), CD3(−), CD10(−)	Right upper abdomen pain Abdomen CT revealed tumor over right iliac fossa	3 years
Chae et al. [14]	75/M	Mantle cell lymphoma	Laparoscopic appendectomy	CD20(+), Pax-5(+), CD43(+), Bcl-1(+), Bcl-2(+), CD3(−), CD5(−), CD10(−), Bcl-6(−)	Left cervical lymphadenopathy. Right lower abdomen pain.	Unknown
Ambrosio et al. [15]	38/F	Nil	Laparoscopic appendectomy	CD3(+), CD5(+), CD20(+), Cyclin D1(+), SOX-11(+)	Right lower abdominal pain and low-grade fever.	>10 months

M: male; F: female; CD: cluster of differentiation; Bcl: B-cell lymphoma; SOX-11: SRY-box transcription factor-11.

## Data Availability

The data underlying this article will be shared upon reasonable request to the corresponding author.

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
