# Peer review of "Mantle Cell Lymphoma Presenting as Acute Abdominal Syndrome: A Rare Case Report and Literature Review"

_healthcare, 2021, doi:10.3390/healthcare9081000_

Round 1

Reviewer 1 Report

Well-written manuscript, it has teaching value for the readers.

Author Response

Response to Reviewer 1

1. Well-written manuscript, it has teaching value for the readers. Author Reply: We are deeply honored by the time and effort you spent in reviewing this manuscript. In reviewing and revising our manuscript, we are motivated to read more and thus learn more from your comment.

Reviewer 2 Report

The manuscript shows a case report of mantle cell lymphoma presenting as an acute abdominal syndrome.

There are some comments.

  1. It would be better to add gross photos, if possible.
  2. It would be better to describe morphologic features of tumor cells in detail.
  3. It would be better to add high-power photos showing the nuclear features of lymphoma cells.
  4. Cyclin D1 expression can be seen in some cases of CLL/SLL (predominantly in proliferation centers), hairy cell leukemia, and myeloma/plasmacytoma, and diffuse large B-cell lymphoma. It would be better to describe differential diagnosis between mantle cell lymphoma and these tumors in the discussion.

Author Response

Response to Reviewer 2

The manuscript shows a case report of mantle cell lymphoma presenting as an acute abdominal syndrome. 

Author Reply: We are deeply honored by the time and effort you spent in reviewing this manuscript. We have revised the manuscript thoroughly according to your suggestions. All changes have been marked in yellow highlight in the revised manuscript. The responses to your comments are below. 

1.  It would be better to add gross photos, if possible Author Reply: Thanks for your suggestion. We have added the gross photography (Supplementary Figure 1) during laparoscopic appendectomy in the revised manuscript. 

 2. It would be better to describe morphologic features of tumor cells in detail. Author Reply: Thanks for your comments. We have revised the manuscript according to your suggestions. Please see Page 3.

Microscopically, marked reactive atypical cell hyperplasia with mild polypmorphism in the mantle zones, variety of cell types of naive pregerminal center cells, starry sky-like appearance with obscure hyalinized vascular proliferation and epithelioid histiocytes with lamina propria, submucosa, penetrating into muscularis layer, serosa and mesenteric fibroadipose were found. These tumor cells were characterized by a rather uniform and monotonous small to intermediate lymphocytic proliferation. These atypical cells demonstrated diffusely positive immunoreactivity for CD45RO, CD20, and Bcl-2. They also presented positive immunostaining for cyclin D1 of atypical lymphoid cells, and negative for pan-CK, EMA, S-100, CD10, and CD23. In addition, the T cell markers of CD3 and CD5 were positive. All of the above findings were consistent with malignant B-cell lymphoma, and the final diagnosis of MCL was made based on the histology features.

3. It would be better to add high-power photos showing the nuclear features of lymphoma cells.

Author Reply: Thanks for your invaluable comments. We have added the high-power photography of the tumor cells (Supplementary Figure 2) in the revised manuscript.

4. Cyclin D1 expression can be seen in some cases of CLL/SLL (predominantly in proliferation centers), hairy cell leukemia, and myeloma/plasmacytoma, and diffuse large B-cell lymphoma. It would be better to describe differential diagnosis between mantle cell lymphoma and these tumors in the discussion.

Author Reply: Thanks for your invaluable comments. We have revised the manuscript and added the associated references according to your suggestions. Please see Page 7, Lines 18-26. Atypical Chronic lymphocytic leukemia (CLL) and small lymphocytic lymphoma (SLL) may also demonstrate positive for cyclin D1 but do not have CCND1 translocations and usually lack SOX11 expression [29]. Rare cases of diffuse large B-cell lymphomas could be positive for cyclin D1, but they frequently lack CCND1 gene translocation and lack immunoreactivity for SOX-11 [30]. Hairy cell leukemia (HCL) may show positive for cyclin D1 and initially be misdiagnosed as MCL. The absence of the CCND1 gene fusion and the strong expression of CD11C, CD25 and CD103 could be found in HCL [31]. MCL and multiple myeloma/plasmacytoma with 11q13 abnormalities typically express Cyclin D1 transcript variants with longer 5′-UTRs [32]. 

References

  1. Gradowski, J. F., Sargent, R. L., Craig, F. E., Cieply, K., Fuhrer, K., Sherer, C., & Swerdlow, S. H. Chronic lymphocytic leukemia/small lymphocytic lymphoma with cyclin D1 positive proliferation centers do not have CCND1 translocations or gains and lack SOX11 expression. American journal of clinical pathology. 2012, 138, 132-139; DOI: 10.1309/AJCPIVKZRMPF93ET.
  2. Hsiao, S. C. ; Cortada, I. R. ; Colomo, L. ; Ye, H. ; Liu, H. ; Kuo, S. Y. ; Lin, S. H.; Kuo, T. U.; Chuang, S. S. SOX11 is useful in differentiating cyclin D1-positive diffuse large B-cell lymphoma from mantle cell lymphoma. Histopathology. 2012; 61, 685-693; DOI:10.1111/j.1365-2559.2012.04260.x.
  3. Zhou, L.; Xu, H., Zhou, J.; Ouyang, B.; Wang, C. A rare case of hairy cell leukemia with co‑expression of CD5 and cyclin D1: A diagnostic pitfall. Molecular and Clinical Oncology. 2020; 13, 74; DOI:10.3892/mco.2020.2142.
  4. Chinen, Y. ; Tsukamoto, T. ; Maegawa-Matsui, S. ; Matsumura-Kimoto, Y. ; Takimoto-Shimomura, T. ; Tanba, K. ; Mizuno, Y.; Fujibayashi, Y,; Kuwahara-Ota, S.; Shimura, Y.; Kobayashi, T.; Horiike, S.; Taniwaki, M.; Kuroda, J. Tumor-specific transcript variants of cyclin D1 in mantle cell lymphoma and multiple myeloma with chromosome 11q13 abnormalities. Experimental hematology. 2020; 84, 45-53. e1; DOI:10.1016/j.exphem.2020.02.004.

Last, we are deeply honored by the time and effort you spent in reviewing this manuscript. In reviewing and revising our manuscript, we are motivated to read more and thus learn more from your criticisms.

This manuscript is a resubmission of an earlier submission. The following is a list of the peer review reports and author responses from that submission.

Round 1

Reviewer 1 Report

Lee et al have written a case report on Mantle cell lymphoma presenting as appendicitis. Although this precise presentation may not be very common, extranodal MCL with GI-involvement is not very rare so the novelty of this case can be questioned. Further, the conclusion that correctly diagnosing malignancies found in the appendix is already well-established, as is MCL/lymphoma diagnostic procedures. Also, several aspects of interest regarding the MCL is missing, such as blood counts, LD value, PS score, B2M, serum electrophoresis/other disease specific laboratory parameters. Further, information regarding adjuvant chemotherapy would be interesting. 

Also, the small review of MCL in the discussion is too short to be valuable and also contains some simplifications, such as that lack of SOX11 is associated with superior prognosis - this is true for non-nodal leukemic patients (who are often SOX11 negative), but not for nodal non-leukemic patients, among whom SOX11 negativity is often associated with TP53-mutations and a more aggressive disease course.

Although the case report is well-written it is not novel and the MCL information therein is not up to date or very informative, all in all limiting the value of this case report.

Reviewer 2 Report

The article presents a case of acute appenditis due to mantle cell lymphoma involving the appendix. The paper is overall well-presented. Although mantle cell lymphoma is a rare neoplam and the involvment of the appendix is indeed rare, however, i don't feel that the case has educational value, since it has a very typical course of diagnosis.

Reviewer 3 Report

The case report part of this paper is clearly and very well written. However, the review of the literature part requires improvement. There needs to be better description of he review of the literature process that took place. This includes search engines used, search terms and methods used, and inclusion and exclusion criteria of the papers used in the review of the literature.

The presentation in the form of a table is clear and concise and is appropriate.